The main stage of recovery after the end-Permian mass extinction: taxonomic rediversification and ecologic reorganization of marine level-bottom communities during the Middle Triassic

Friesenbichler Evelyn evelyn.friesenbichler@pim.uzh.ch
Hautmann Michael
Bucher Hugo
Paläontologisches Institut und Museum, University of Zurich , Zurich , Switzerland
Hedrick Brandon
Electronic publication date: 2021 Jul 20
Publication date: 2021
Volume: 9
Electronic Location ID: e11654
Received 2020 Oct 29; Accepted 2021 May 31
Copyright: ©2021 Friesenbichler et al.
Copyright year: 2021
Copyright holder: Friesenbichler et al.
License: This is an open access article distributed under the terms of the Creative Commons Attribution License, which permits unrestricted use, distribution, reproduction and adaptation in any medium and for any purpose provided that it is properly attributed. For attribution, the original author(s), title, publication source (PeerJ) and either DOI or URL of the article must be cited.
License URL: https://creativecommons.org/licenses/by/4.0/

Keywords: Biotic recovery, Diversification, Palaeoecology, Middle Triassic, End-Permian mass extinction

Funding: Swiss National Science Foundation 200021-162402/1 This work was supported by the Swiss National Science Foundation (grant 200021-162402/1 to Michael Hautmann). The funders had no role in study design, data collection and analysis, decision to publish, or preparation of the manuscript.

==============================
The recovery of marine life from the end-Permian mass extinction event provides a test-case for biodiversification models in general, but few studies have addressed this episode in its full length and ecological context. This study analyses the recovery of marine level-bottom communities from the end-Permian mass extinction event over a period of 15 Ma, with a main focus on the previously neglected main phase during the Middle Triassic. Our analyses are based on faunas from 37 lithological units representing different environmental settings, ranging from lagoons to inner, mid- and outer ramps. Our dataset comprises 1562 species, which belong to 13 higher taxa and 12 ecological guilds. The diversification pattern of most taxa and guilds shows an initial Early Triassic lag phase that is followed by a hyperbolic diversity increase during the Bithynian (early middle Anisian) and became damped later in the Middle Triassic. The hyperbolic diversity increase is not predicted by models that suggest environmental causes for the initial lag phase. We therefore advocate a model in which diversification is primarily driven by the intensity of biotic interactions. Accordingly, the Early Triassic lag phase represents the time when the reduced species richness in the wake of the end-Permian mass extinction was insufficient for stimulating major diversifications, whereas the Anisian main diversification event started when self-accelerating processes became effective and stopped when niche-crowding prevented further diversification. Biotic interactions that might drive this pattern include interspecific competition but also habitat construction, ecosystem engineering and new options for trophic relationships. The latter factors are discussed in the context of the resurgence of large carbonate platforms, which occurred simultaneously with the diversification of benthic communities. These did not only provide new hardground habitats for a variety of epifaunal taxa, but also new options for grazing gastropods that supposedly fed from microalgae growing on dasycladaceans and other macroalgae. Whereas we do not claim that changing environmental conditions were generally unimportant for the recovery of marine level-bottom communities, we note that their actual role can only be assessed when tested against predictions of the biotic model.

Introduction

The end-Permian mass extinction was the most severe extinction event in Earth’s history and witnessed the extinction of 81% (Stanley, 2016) to 96% (Raup, 1979) of marine species. The aftermath of this event provides a unique test case to study how life rediversified after such a massive loss of species richness and which evolutionary processes were involved. In this study, these processes were analyzed in the ecological context in which they occurred. Previous studies with a similar scope concentrated on the Early Triassic (e.g., Schubert & Bottjer, 1995; Twitchett & Wignall, 1996; Hofmann, Hautmann & Bucher, 2013; Hofmann et al., 2013; Hofmann, Hautmann & Bucher, 2015; Foster et al., 2015; Hautmann et al., 2015; Foster et al., 2017; Foster, Danise & Twitchett, 2017), the time interval of ca. 5 Ma after the end-Permian mass extinction. These studies revealed, among other things, that (1) the recovery of the nekton (e.g., ammonoids and conodonts; Orchard, 2007; Brayard et al., 2009; Brayard & Bucher, 2015) is remarkably different from that of the benthos (Hofmann, Hautmann & Bucher, 2013; Hofmann et al., 2013; Hofmann et al., 2014; Friesenbichler et al., 2019), (2) the main recovery phase of the benthos did not start before the Middle Triassic (Nützel, 2005; Hautmann, 2007; Hofmann et al., 2014; Friesenbichler et al., 2021), (3) benthic ecosystem recovery was not completed before the end of the Middle Triassic (Hausmann & Nützel, 2015) and (4) the resurgence of large carbonate platforms and the recovery of reefs during the Middle Triassic (e.g., Gaetani et al., 1981; Senowbari-Daryan et al., 1993; Flügel, 2002) possibly played an important role in the shift of taxonomic composition of benthic communities (Friesenbichler et al., 2019). Despite the fact that the Middle Triassic was obviously a crucial time for the recovery of benthic marine communities, corresponding studies (e.g., Payne, 2005; Payne et al., 2006a; Payne et al., 2006b; Song et al., 2011; Velledits et al., 2011; Foster & Sebe, 2017; Friesenbichler et al., 2019; Friesenbichler et al., 2021) are still scarce. Relevant questions in the ecological-evolutionary context include: Was the delay in benthic recovery caused by adverse environmental conditions during the Early Triassic (e.g., Hallam, 1991; Wignall & Twitchett, 1996; Pruss & Bottjer, 2004; Song et al., 2014), or is it an intrinsic aspect of the diversification dynamics in a largely vacated ecospace (Hautmann et al., 2015)? How did community structures, ecological guilds and ecosystems change during the Middle Triassic, and how did these changes translate into observed diversification patterns? Did the resurgence of large carbonate platforms affect the guild structure, and if yes, which evolutionary processes were involved in the colonization of this new habitat type? What was the relative role of biotic processes such as interspecific competition, habitat alteration, ecosystem engineering and niche construction versus abiotic environmental conditions as controlling factors in diversification and ecosystem evolution? To which diversification model do the empirical patterns conform, and which conclusions does this imply for the diversification of life in general?

Materials & Methods

This study is based on data of macroinvertebrate occurrences compiled at the species level from palaeontological studies on Early and Middle Triassic lithological units (Data S1). We preferred this approach over using data from the Palaeobiology Database because it allowed us to control the correctness of the data. The evaluated studies include primarily monographs with well documented faunal lists, but we also considered studies that provide detailed information about macroinvertebrate occurrences including taxonomic assignments. Prime criteria for our selection of studies are the quality of documentation and the coverage of time intervals and environments. Moreover, we sought to include the most diverse lithological unit of each time interval as representatives for the maximum diversity that was possible at a given stage of recovery (see below).

We did not include any species for which we haven’t studied the original description and figures. Taxa reported from thin sections or polished slabs were not considered because taxonomic assignments of these taxa were in most cases difficult to verify. The complete dataset includes 37 lithological units comprising 1562 species belonging to gastropods (695 species), bivalves (587 species), brachiopods (147 species), crinoids (33 species), echinoids (31 species), crustaceans (25 species), ophiuroids (12 species), scaphopods (nine species), serpulids (eight species), microconchids (six species), asteroids (four species), bryozoans (four species) and polyplacophores (one species; Data S1). Our approach led to a certain overemphasis of European lithological units (27) in comparison to those from Asia (five) and America (five units from the USA), but the prevailing cosmopolitism of marine faunas of this time (Kristan-Tollmann & Tollmann, 1982; Schubert & Bottjer, 1995) justifies the priority of the quality of the documentation over geographic coverage.

Data preparation

The investigated lithological units represent different environments (Data S1). Investigated environments include lagoons (including carbonate platform interiors), inner ramps (above the fair-weather wave base), mid-ramps (between the fair-weather wave base and the storm wave base) and outer ramps (below the storm wave base). The assignment of faunas to a particular environment is based on the information given by the original authors or inferred from the geological and sedimentological context. Studies on lithological units that represent a range of different environments were only considered if the distribution of species within these environments was clear. Therefore, we did not include the Ladinian part of the Cassian Formation (Italy; Urlichs, 2017) in our analyses, although it probably represents the highest benthic diversity of this time interval.

From the collected studies, fossil lists were compiled for each lithological unit. If information about one lithological unit was taken from more than one study, attention was paid on possible synonyms.

Each species was assigned to an ecological guild, usually based on information given by the original authors. In cases where such information was missing, the ecology was inferred from functional morphology, mode of life of Recent species and information from the literature. Species included in this study represent shallow to moderately deep infaunal suspension-feeders, shallow infaunal deposit-feeders, deep infaunal suspension-feeders, endobyssate suspension-feeders, epibyssate suspension-feeders, free-lying suspension-feeders, cemented epifaunal suspension-feeders, pedunculate suspension-feeders, epifaunal herbivores and/or detritus feeders, epifaunal carnivores, erected epifaunal suspension-feeders and epifaunal detrivore-suspension-feeders (Table 1). Gastropods, echinoids and ophiuroids were collectively assigned to epifaunal herbivores and/or detritus-feeders and epifaunal detrivore-suspension-feeders, respectively. We are aware that this might be an over-generalization, but the shell morphology of these taxa does not provide evidence for more specialized feeding in extinct species, and the ecology of Recent species justifies this assignment for most species. Species with uncertain ecology were considered for calculating species richness but were not included in the ecological analyses.

Table 1 List of ecological guilds and representing taxa.

Ecological guild	Taxa	
Shallow to moderately deep infaunal suspension-feeders	Bivalves, inarticulate brachiopods	
Shallow infaunal deposit-feeders	Bivalves, scaphopods	
Deep infaunal suspension-feeders	Bivalves	
Endobyssate suspension-feeders	Bivalves	
Epibyssate suspension-feeders	Bivalves	
Free-lying epifaunal suspension-feeders	Bivalves	
Cemented epifaunal suspension-feeders	Bivalves, serpulids, inarticulate brachiopods, bryozoans, microconchids	
Pedunculate suspension-feeders	Articulate brachiopods	
Epifaunal herbivores and/or detritus-feeders	Gastropods, echinoids, polyplacophors	
Epifaunal carnivores	Asteroids, crustaceans	
Erected epifaunal suspension-feeders	Crinoids	
Epifaunal detrivore-suspension-feeders	Ophiuroids	

Time resolution is provided at the substage level. For lithological units that extend across substages, we assumed constant diversity across substages if the literature suggests that this assumption is justified (e.g., data from the Germanic basin; Schmidt, 1928; Schmidt, 1938).

Data analyses

Conventional diversity studies (e.g., Sepkoski, 1984; Sepkoski, 1997) through deep time have relied on higher taxonomic levels (genera, families, orders) as surrogates for the species level; however this approach may cause a strong bias (Benton, 2001). Unfortunately, compiling reliable global species level curves is virtually impossible because of the higher incompleteness of the fossil record for species than for higher taxa and the necessity of eliminating synonyms. We suggest here a feasible solution for circumventing these problems. We use the maximum species richness among all lithological units in a given time interval as a surrogate for the potentially highest recovery stage that biosphere could reach at this time point after the end-Permian mass extinction (Figs. 1 and 2). In other words, for each substage and ecological guild, the lithological unit that contains the highest number of species was used as a reference point. Using this approach, synonyms are not a major issue because it is irrelevant for the diversity of a given lithological unit whether the same species is described from another lithological unit under a different name.

Figure 1 Maximum species richness throughout the Early and Middle Triassic.

The Longobardian is palish to indicate that the decrease of species richness is not a primary signal. Light grey bars on top indicate the number of lithological units representing each time interval. In cases where information does not come from all lithological units representing the corresponding time interval, the actual number of lithological units from which information comes from is indicated by dark grey bars. Abbreviations: A, Aegean; B, Bithynian; D, Dienerian; Fassan, Fassanian; G, Griesbachian; I, Induan; Illyr, Illyrian; Longobard, Longobardian; Pel, Pelsonian; Smith, Smithian. Absolute ages according to Cohen et al., (2013, updated) and Ovtcharova et al. (2015). Information about the relative duration of substages come from Nawrocki & Szulc (2000), Götz, Szulc & Feist-Burkhardt (2005), Galfetti et al. (2007), Ovtcharova et al. (2006) and Widmann et al. (2020). Symbols for ecological guilds redrawn after Aberhan (1994). See text for further explanation.

Figure 2 Maximum species richness in different environments.

(A) lagoons, (B) inner ramps, (C) mid-ramps and (D) outer ramps. The Longobardian is palish to indicate that the decrease of species richness is not a primary signal. Light grey bars on top indicate the number of lithological units representing each time interval. In cases where information does not come from all lithological units representing the corresponding time interval, the actual number of lithological units from which information comes from is indicated by dark grey bars. Due to our data selection, the following guilds are not shown in this figure although reported in polished slabs: shallow infaunal deposit-feeders in late Griesbachian-Smithian and middle Spathian inner ramps (one species, respectively) and Smithian mid-ramps (one species), cemented epifaunal suspension-feeders in early-middle Spathian inner ramps (one species) and Dienerian-Smithian outer ramps (two species), epifaunal herbivores and/or detritus-feeders in early-middle Spathian (two species) and late Pelsonian-early Illyrian (five species) outer ramps. Abbreviations: Spath, Spathian. Further abbreviations, symbols for ecological guilds and information about the relative duration of substages as in Fig. 1. See text for further explanation.

Furthermore, we investigated relative changes in the ecological composition throughout the Early and Middle Triassic (Figs. 3 and 4). For this purpose, the guild-species diversity (number of species per guild) for each substage was calculated. To do so, we summarized the number of species that occurred in all lithological units per substage. The sum of species was corrected for double counts, meaning that a species that occurred in several lithological units per substage was only counted once.

Figure 3 Guild-species diversity throughout the Early and Middle Triassic.

Bars on top indicate the number of lithological units representing each time interval. Abbreviations, symbols for ecological guilds and information about the relative duration of substages as in Fig. 1. See text for further explanation.

Figure 4 Guild-species diversity of different environments.

(A) lagoons, (B) inner ramps, (C) mid-ramps and (D) outer ramps. Bars on top indicate the number of lithological units representing each time interval. Due to our data selection, the following guilds are not shown in this figure although reported in polished slabs: shallow infaunal deposit-feeders in late Griesbachian-Smithian and middle Spathian inner ramps (one species, respectively) and Smithian mid-ramps (one species), cemented epifaunal suspension-feeders in early-middle Spathian inner ramps (one species) and Dienerian-Smithian outer ramps (two species), epifaunal herbivores and/or detritus-feeders in early-middle Spathian (two species) and late Pelsonian-early Illyrian (five species) outer ramps. Abbreviations as in Figs. 1 and 2. Symbols for ecological guilds and information about the relative duration of substages as in Fig. 1. See text for further explanation.

Results

Figure 1 shows the huge differences in species richness between Early and Middle Triassic benthic communities that hold across different environments (Fig. 2). The apparent decrease of species richness in the Longobardian (late Ladinian; Figs. 1 and 2) is an artefact of the scarcity of data that would disappear e.g., if time-resolved data from the Ladinian part of the Cassian Formation were available (see Methods). Currently, only one lithological unit represents the middle and late Longobardian in Figs. 1 and 2. Scarce data may also underlie changes in guild-species diversity seen in the Smithian (early Olenekian) and the Aegean (early Anisian; Fig. 3).

Figure 5 Maximum species richness throughout the Early and Middle Triassic of ecological guilds.

(A–B) epifaunal herbivores and/or detritus-feeders, (C–D) shallow to moderately deep infaunal suspension-feeders, endobyssate suspension-feeders, epibyssate suspension-feeders and pedunculate suspension-feeders, (E-F) shallow infaunal deposit-feeders, deep infaunal suspension-feeders, free-lying suspension-feeders and cemented microcarnivores, (G-H) cemented epifaunal suspension-feeders, erected epifaunal suspension-feeders, epifaunal carnivores and epifaunal-detrivore-suspension-feeders. Number of species illustrated with linear scale (A, C, E and G) and logarithmic scale (B, D, F and H). Filled symbols represent faunas that are more or as diverse as older ones, whereas empty symbols represent faunas that are less diverse than older ones. Abbreviations as in Fig. 1 and Fig. 2. Information about the relative duration of substages as in Fig. 1. See text for further explanation.

Epifaunal herbivores and/or detritus feeders (i.e., gastropods) diversified most quickly in the Middle Triassic. Shallow to moderately deep and deep infaunal, endobyssate, epibyssate, free-lying, cemented and pedunculate suspension-feeders as well as shallow infaunal deposit-feeders started diversifying at the same time (Fig. 5) but their species richness levelled out at lower plateaus.

Bivalves, gastropods and brachiopods were the most diverse taxa in Early and Middle Triassic benthic communities (Data S1). Bivalves had a relatively low and constant species richness throughout the Early Triassic and diversified quickly in the Bithynian (early middle Anisian) followed by a plateau (Figs. 6A and 7). Except for the Smithian Sinbad Formation, the species richness of gastropods was low in the Early Triassic and started to increase rapidly in the Bithynian and their species richness started to exceed that of bivalves from the Pelsonian (late middle Anisian) onwards (Figs. 6B and 7). Brachiopods also diversified in the Bithynian but not to the same extent as bivalves and gastropods did, and their species richness maintained a Middle Triassic plateau (Fig. 6C and Fig. 7). Especially gastropods but also brachiopods were relatively diverse in lithological units that are associated with carbonate platforms (red lines in Figs. 6B and 6C). Figure 6D shows a change in taxonomic composition from bivalve-dominated Early Triassic to increasingly more gastropod-dominated Middle Triassic faunas. Almost all lithological units associated with carbonate platforms are dominated by gastropods, which was already noticed by Friesenbichler et al. (2019).

The guild-species diversity of epifaunal herbivores and/or detritus-feeders (i.e., gastropods) changed strongest among all ecological guilds during the studied time interval. During the Early Triassic they were the third most diverse guild but they started to diversify quickly during the Bithynian and became almost instantaneously the dominating guild in Middle Triassic benthic communities (Fig. 3). The only environment where this trend is not observed are outer ramps, where their species richness remained constant (Figs. 2D and 4D). Figures 5A and 5B show that their rapid diversification started in the Bithynian and continued throughout the Anisian.

Epibyssate suspension-feeders (chiefly pteriomorphian bivalves) were the most diverse guild in the Induan and the Spathian (late Olenekian). During the Bithynian their absolute richness increased whereas their relative richness decreased (Figs. 1 and 3, 5C, 5D) and continued to decrease slightly throughout the Middle Triassic. This decrease is also apparent in lagoons, inner and mid-ramps, whereas in outer ramps their relative diversity increased slightly (Fig. 4).

Figure 6 Species richness of the most diverse taxa and the gastropod/bivalve-ratio throughout the Early and Middle Triassic.

(A) bivalves, (B) gastropods, (C) brachiopods and (D) the gastropod/bivalve-ratio. Each line represents one lithological unit. Red lines mark lithological units and ratios associated with carbonate platforms. Abbreviations as in Fig. 1 and Fig. 2. Information about the relative duration of substages as in Fig. 1. (A) modified after Friesenbichler et al. (2021) and (D) modified after Friesenbichler et al. (2019). See text for further explanation.

Figure 7 Maximum species richness of bivalves, gastropods and brachiopods throughout the Early and Middle Triassic.

(A–B) in general and (C–D) in mid-ramps. Number of species illustrated with linear scale (A and C) and logarithmic scale (B and D). Filled symbols represent faunas that are more or as diverse as older ones, whereas empty symbols represent faunas that are less diverse than older ones. Abbreviations as in Fig. 1 and Fig. 2. Information about the relative duration of substages as in Fig. 1. See text for further explanation.

Shallow to moderately deep infaunal suspension-feeders (bivalves and inarticulate brachiopods) were among the most diverse guilds during the Early Triassic but their relative richness decreased gradually after the Aegean (Figs. 1 and 3). The same trend occurred in the guild-species diversity in lagoons, mid- and outer ramps; however, in mid-ramps the relative species richness of shallow to moderately deep infaunal suspension-feeders was higher in the late Illyrian (late Anisian) to late Fassanian (early Ladinian) interval than in the late Pelsonian to early Illyrian (Figs. 4A, 4C and 4D). In inner ramp settings, the relative amount of shallow to moderately deep infaunal suspension-feeders decreased already during the Spathian. Their main recovery phase started in the Bithynian but the diversification rate seemed to slow down afterwards (Figs. 5C and 5D).

At the beginning of the Spathian, the relative richness of endobyssate suspension-feeders (bivalves) started to increase, but after the Aegean, it decreased continuously (Fig. 3). The same trend occurred in lagoons and outer ramps. In inner and mid-ramp settings, the relative amount of endobyssate suspension-feeders was more or less constant after the Spathian (Fig. 4). The species richness of endobyssate suspension-feeders increased constantly during the Early Triassic, followed by a rapid rise in species richness in the Bithynian. However, after this increase their species richness remained constant (Figs. 5C and 5D).

The relative amount of pedunculate suspension-feeders (articulate brachiopods) fluctuated. They were relatively diverse during the Griesbachian (early Induan) and after the Bithynian but in between they were rather uncommon. After the early Illyrian they became gradually less diverse (Fig. 1). In Early Triassic mid-ramp settings, they are only recorded from the Griesbachian. Their relative importance was high during the Bithynian, but afterwards they became less common. In outer ramps from the Early Triassic, this guild is only reported from the Griesbachian and Dienerian (late Induan); however, in Middle Triassic outer ramps pedunculate suspension-feeders were the most diverse guild (Fig. 4D). They experienced a rapid rise in diversification in the Bithynian (Figs. 5C and 5D).

Other guilds were rather uncommon during the Early and Middle Triassic or not represented in all time slices (Figs. 1 and 3). Deep infaunal suspension-feeders (siphonate bivalves) are first reported from the Spathian (Fig. 3) and showed a weak increase in species richness in the Bithynian (Figs. 5E and 5F). The species richness of shallow infaunal deposit-feeders (i.e., nuculid bivalves and scaphopods) and free-lying epifaunal suspension-feeders (bivalves) increased in the Bithynian but stayed constant during the remaining of the Middle Triassic (Figs. 5E and 5F). Cemented epifaunal suspension-feeders (e.g., oysters, serpulids, microconchids) and erected epifaunal suspension-feeders (i.e., crinoids) also started to diversify during the Bithynian, but their later diversification pattern is poorly documented (Figs. 5G and 5H). Epifaunal detrivore-suspension feeders (i.e., ophiuroids) and epifaunal carnivores (i.e., asteroids and crustaceans) started to diversify in the Aegean or Bithynian (Figs. 5G and 5H).

A remarkable aspect of the guild analysis is that the explosive Anisian diversity increase was not associated with the evolution of new benthic guilds, confirming previous observations by Foster & Twitchett (2014).

Discussion

Possible biases

Our data indicate an explosive increase in benthic diversity at the beginning of the Middle Triassic that followed an extended Early Triassic lag phase (Fig. 1). Can this pattern result from a bias in preservation, rock exposure or research history?

A preservation bias in Early Triassic strata has often been proposed as a possible reason for the generally low species richness at that time (e.g., Erwin, 1996; Wignall & Benton, 1999; Peters & Foote, 2002; Nützel & Schulbert, 2005; Hautmann et al., 2011), but well preserved Early Triassic benthic faunas have been reported worldwide, e.g., from the western USA (Batten & Stokes, 1986; Hautmann & Nützel, 2005; Nützel & Schulbert, 2005; Pruss, Payne & Westacott, 2015; Brayard et al., 2017), South China (Kaim et al., 2010; Hautmann et al., 2011; Hautmann et al., 2015; Foster et al., 2019), Russia (Shigeta et al., 2009) and Pakistan (Wasmer et al., 2012; Kaim et al., 2013); even silicified faunas are known from the Early Triassic of Oman (Twitchett et al., 2004; Wheeley & Twitchett, 2005; Oji & Twitchett, 2015) and Svalbard (Foster, Danise & Twitchett, 2017). However, all these well preserved Early Triassic benthic communities are much less diverse than communities from comparable settings in the Middle Triassic, so their low diversity is probably a primary signal.

Except for taxonomic oversplitting and synonyms (see Methods), the history of research and the size of exposure influence our knowledge on species richness. In our data, the most extreme potential bias in this respect is between the stratigraphically adjacent data points from the Bithynian Tubiphytes-Limestone Member (Romania) and the Jena Formation (Germany), which corresponds to the strongest increase in diversity within the studied time interval. The described fauna from the Tubiphytes-Limestone Member comes from only 1.5 m3 of rock material (Gradinaru & Gaetani, 2017; Nützel, Kaim & Grădinaru, 2018; Friesenbichler et al., 2021), and rarefaction analysis indicates that the bivalve fauna is actually insufficiently sampled (Friesenbichler et al., 2021). In contrast, the Germanic Triassic looks back on a long history of research and is represented by a large outcrop area, which should correspond to a significantly higher completeness of sampled biodiversity. However, correcting the data point from the Tubiphytes-Limestone Member towards a higher diversity would solely shift the starting point of the main diversity increase from the middle to the early Bithynian, but not alter the overall shape of the diversity trajectory.

Extrinsic or intrinsic control?

The observation of an Early Triassic lag phase in the recovery from the end-Permian mass extinction is not new. Schubert & Bottjer (1995) were among the first to demonstrate this delay in rediversification based on regional data of post-extinction communities in the western USA. They suggested that the “long pre-radiation period” could be due (1) to the extraordinarily magnitude of the end-Permian mass extinction and/or (2) the persistence of environmental stress. The first explanation implies an evolutionary slow-down of diversification rates that correlates with the extinction magnitude, possibly because there is a positive feedback between species richness and rates of speciation, which becomes ineffective if diversity falls below a critical threshold (Solé et al., 2010; Hautmann et al., 2015). The second (environmental) explanation has found disproportionally more attention in the literature, which offers manifold scenarios of poisonous, acidic, oxygen deficient and lethally hot oceans that prevented life from re-diversifying after the great dying at the end of the Permian (e.g., Wignall & Twitchett, 1996; Payne et al., 2007; Joachimski et al., 2012; Sun et al., 2012; Song et al., 2014). Support for the second explanation has been sought in the demonstration of adverse environmental conditions during the Early Triassic from geochemical or palaeontological proxies (e.g., Payne et al., 2010; Sanei, Grasby & Beauchamp, 2012; Schobben et al., 2014; Tian et al., 2014; Rothman et al., 2014; Wei et al., 2015), but the idea of pervasive hostile conditions in the Early Triassic has also been criticized for being at variance with a variety of observations. These include: (1) benthic communities from various environmental settings and palaeogeographically distant regions show little indications for unusual environmental stress (Twitchett et al., 2004; Hautmann et al., 2011; Hautmann et al., 2015; Hofmann, Hautmann & Bucher, 2013; Hofmann et al., 2013; Hofmann et al., 2014); (2) ichnofaunas were diverse and complex soon after the extinction event in different palaeolatitudes (Beatty, Zonneveld & Henderson, 2008; Hofmann et al., 2011); (3) shell sizes of many Early Triassic gastropods reached large sizes (Brayard et al., 2010; Brayard et al., 2011a; Brayard et al., 2015); (4) the diversification of ammonoids was extremely rapid and displayed a cyclic pattern in time (Brayard et al., 2009; Brayard & Bucher, 2015). (5) biomass productivity of marine benthos was high in spite of low taxonomic diversity (Brosse et al., 2019) and (6) marine apex predators were present throughout the Early Triassic (Scheyer et al., 2014). This criticism does not deny the existence of environmental stress in the Early Triassic, but it suggests that it was locally and temporarily restricted and had differential impacts on the benthos and nekton.

Another class of explanations assumes that a breakdown of primary production in the Early Triassic caused a collapse of the food pyramid. According to these models, the low diversity in the Early Triassic corresponds to a time of reduced primary production (e.g., Grasby et al., 2020) and the recovery of marine life required a successive rebuilding of trophic levels (Chen & Benton, 2012, p. 379). However, Chen & Benton (2012) suggested that this model predicts a logistic rediversification curve, which would be at variance with the extended Early Triassic lag phase. Palaeontological data also demonstrate the presence of apex predators in the Early Triassic (Scheyer et al., 2014), which indicates that trophic levels between the base and the top of the food pyramid were at least partly intact. We add here that low primary production per se does not provide an explanation for low diversity. Theory rather predicts that selection for fitness in resource-poor environments favours specialized, efficient populations and therefore results in high diversity (Valentine, 1971). The high diversity of Recent coral reefs and deep-sea communities (Grassle & Maciolek, 1992; Veron et al., 2009) are well-known examples that conform to this prediction.

In contrast to the idea of environmentally driven delay (EDD) models, the biotic interaction (BI) model does not only provide an explanation for the delayed recovery but it also makes a prediction for the diversification pattern that followed the initial lag phase. Mathematically, the BI model can be written as a differential equation that can be solved numerically. It contains a hyperbolic term that accounts for the positive feedback of species interactions on rates of diversification, which is usually combined with a damping term that limits the increase when the effects of niche pre-emption and ecological crowding become dominant (Solé et al., 2010, equation 10; Hautmann et al., 2015, equation 1). In contrast to the more familiar logistic model (Sepkoski, 1978; 1984), the shape of the resulting curve is decidedly asymmetrical (compare Figs. 8A and 8C). It displays an extended early (left) branch, which represents the prolonged lag phase when feedback processes were ineffective due to low species richness, followed by an explosive increase that stops relatively suddenly when ecospace is filled (Fig. 8C). However, the exact shape of the curve depends upon two parameters: (1) the starting diversity that can be expressed as percentage of the carrying capacity or as the percentage of surviving species, and (2) the proportionality factor, which represents the feedback strength. Depending on these two parameters, which are insufficiently known, the shape of the hyperbolic damped diversification curve can appear relatively similar to a logistic shape under certain conditions, but a qualitative difference between the two models always remains. This difference can be visualized in semilogarithmic plots, where a hyperbolic curve is reflected by an initial increase in the slope of the diversification curve, whereas a logistic curve is represented by a decreasing slope (Figs. 8B and 8D).

Figure 8 Diversification curves.

(A–B) Logistic with number of species illustrated with linear scale (A) and logarithmic scale (B). (C–D) Hyperbolic-damped with number of species illustrated with linear scale (C) and logarithmic scale (D).

The majority of the semilogarithmic plots presented in this study (Figs. 5 and 7) is conform to the hyperbolic-damped (= BI) model. This applies to the diversification curves of bivalves, gastropods and brachiopods in general, bivalves from mid-ramp settings, as well as shallow infaunal deposit-feeders (i.e., nuculid bivalves and scaphopods) and free-lying epifaunal suspension feeders (some bivalves), with their sudden increase in species richness during the Bithynian that is followed by a plateau, which indicates a hyperbolic-damped increase in species richness. Likewise, the diversification trends of gastropods in mid-ramp settings, epibyssate suspension-feeders (some bivalves), epifaunal herbivores and/or detritus-feeders (i.e., gastropods), erected epifaunal suspension-feeders (i.e., crinoids) and pedunculate suspension-feeders (i.e., articulate brachiopods) show a sudden increase in species richness in the Bithynian, which is as indicative of a hyperbolic trajectory. The data from some other guilds do not provide an unequivocal support for either the hyperbolic or the logistic model. This concerns shallow to moderately deep infaunal, endobyssate and cemented epifaunal suspension-feeders (e.g., some bivalves, serpulids) as well as epifaunal detrivore-suspension-feeders (i.e., ophiuroids). The remaining taxa and guilds (i.e., brachiopods of mid-ramp settings, deep infaunal suspension-feeders and epifaunal carnivores like asteroids and crustaceans) do not match with any model, possibly because of their scarcity in our dataset.

A conclusion from these observations is that they suggest a strong effect of biotic interactions on rates of diversification. This is the prime difference to the standard logistic model, which makes a neutral assumption with respect to biotic interactions. In the logistic model, the initially exponential increase is solely an effect of an increasing number of species that each have constant diversification rates. This assumption is clearly incompatible with the explosive diversification revealed in our dataset (Fig. 1). Notably, this is not an isolated observation. Miller & Sepkoski (1988) found “hyperexponential bursts” in the Phanerozoic diversity curve of bivalves during the Ordovician diversification and following the end-Permian and end-Cretaceous mass extinctions, which likely represent times of hyperbolic increase. Markov & Korotayev (2007) even proposed that the Phanerozoic biodiversity curve from which Sepkoski (1978); Sepkoski, 1984) derived his model is actually better described by a hyperbolic model.

Implicit in these finds is that we cannot expect an early rediversification if the feedback mechanism between richness and rates of diversification was ineffective after an extreme diversity crash, whether or not the environment was favourable. Does this conclusion refute the EDD model? It partly does, because it means that at least a portion of the lag phase is always attributed to BI. It is therefore incorrect to ascribe the full length of the lag phase to EDD. However, it is currently also uncertain whether the full length of the Early Triassic lag phase can be explained solely by the BI model. The length of the lag phase in this model depends upon two parameters, the starting diversity and the proportionality factor, which are not precisely known. It is therefore possible that adverse environmental conditions delayed or interrupted the early recovery whereas the BI model explains a subsequent extension of the lag phase. Clarifying the exact parameters in the BI model equation for different taxa is required for estimating the length of the intrinsic delay and thus the potential for EDD as an additional explanation for the Early Triassic lag phase.

From an evolutionary viewpoint, the most relevant aspect is the nature of biotic interactions that cause positive feedbacks between species richness and rates of diversification. Solé et al. (2010) were vague with respect to possible feedback processes, suggesting that increasing numbers of ecological interactions might provide the context for new opportunities to speciate, but they also noted that this process might operate too slowly to explain explosive diversification. Alternatively, they proposed that the growing number of potential interactions might drive the increase in the number of species. Hautmann et al. (2015) reconsidered the problem and identified interspecific competition as a potential main driver of hyperbolic-damped diversification curves. Accordingly, promotion of niche differentiation is the dominant outcome of interspecific competition until a critical level of ecological saturation has been reached, at which further division of niche space requires increasingly elaborated adaptations that are more and more unlikely to evolve. Above such a saturation limit, outcompeting of existing species becomes the prevalent effect of competition, which damps further diversification. A strength of this model is that it correctly predicts different diversification trajectories for taxa with supposedly different intensities of interspecific competition, for which slowly recovering bivalves contrasted by explosively rediversifying ammonoids have been cited as an example (Hautmann et al., 2015). The fact that the phase of the hyperbolic diversity increase was not associated with the origin of new guilds further supports the competition-driven scenario, because competition is most intense between species with similar lifestyles. However, Hautmann et al. (2015) also noted that many other aspects in addition to competition affect actual patterns of diversification. In the following, we discuss some possible additional factors that were linked to the resurgence of carbonate platforms in the Middle Triassic.

Habitat construction and biotic interactions on resurging large carbonate platforms

The recovery of large carbonate platforms after the end-Permian mass extinction started in the Anisian (early Middle Triassic; Gaetani et al., 1981; Senowbari-Daryan et al., 1993; Berra, Rettori & Bassi, 2005), and coincided with the main diversification of most benthic marine invertebrate taxa (Figs. 5 and 7). One of the oldest Triassic carbonate platforms is represented by the Tubiphytes-Limestone Member of the Caerace Formation in Romania. This Bithynian biostrome is dominated by the micro-encruster Tubiphytes and large volumes of synsedimentary cements (Popa, Panaiotu & Graˇdinaru, 2014) and provided the basis for a rich hardground community that is much more diverse than contemporaneaus level-bottom faunas (Gradinaru & Gaetani, 2017; Forel & Grădinaru, 2018; Nützel, Kaim & Grădinaru, 2018; Friesenbichler et al., 2021). Its high amount of newly described species is potentially related to the new habitat type. This applies to the dominant taxa, i.e., epibyssate bivalves, gastropods and brachiopods (Data S1) that were well adapted to the hardgrounds provided by the Tubiphytes-microbial buildup (Friesenbichler et al., 2021). Further Middle Triassic Tubiphytes-dominated buildups are known from the Iranian carbonate mounds of Nakhlak (late Bithynian; Berra et al., 2012), the Italian Camorelli platform (Bithynian-Pelsonian; Gaetani & Gorza, 1989) and Dont Formation (Pelsonian-Illyrian; Blendinger, 1983; Fois & Gaetani, 1984), the Hungarian Aggtelek reef (Pelsonian-Ladinian; Velledits et al., 2011; Velledits, Hips & Pero, 2012) and Chinese reefs (Anisian; Enos, Wei & Yan, 1997; Lehrmann, 1999; Enos et al., 2006; Payne et al., 2006a; Payne et al., 2006b; Lehrmann et al., 2007). The latter flourished on the Great Bank of Guizhou, an isolated carbonate platform in the Nanpanjiang Basin, and represent the oldest Triassic platform margin reefs (Lehrmann, Wei & Enos, 1998; Payne et al., 2006a; Payne et al., 2006b).

Reefs existed throughout the Triassic, but their composition changed stepwise (see Martindale, Foster & Velledits, 2019). The oldest Triassic reefs are represented by microbial-metazoan reefs that formed immediately after the end-Permian mass extinction (e.g., Lehrmann et al., 2001; Flügel, 2002; Pruss et al., 2006; Wu et al., 2007; Kiessling & Simpson, 2011; Kershaw et al., 2011; Ezaki, Liu & Adachi, 2012; Yang et al., 2015; Friesenbichler et al., 2018). The first metazoan reefs are reported from the Olenekian (late Early Triassic) and represent small sponge biostromes and bivalve build-ups (Pruss, Payne & Bottjer, 2007; Brayard et al., 2011b; Marenco et al., 2012). Reefs developed quickly in the middle Anisian (Flügel, 2002) and in addition to Tubiphytes other organisms such as sphinctozoans (segmented calcareous sponges), dasycladacean algae, corals, and in some cases also bivalves became important reef-builders (Flügel, 2002; Fürsich & Hautmann, 2005; Senowbari-Daryan & Link, 2011). These organisms acted as ecological engineers and created new niches that provided opportunities for additional species. Well-documented examples are bivalve-crinoid reef mounds from the Trochitenkalk Formation (middle Illyrian, Germany; e.g., Hagdorn, 1978; Sellwood & Fürsich, 1981; Hagdorn & Mundlos, 1982; Flügel, 2002; Hagdorn, 2004; Hagdorn & Nitsch, 2009), where shell beds were incrusted by cementing bivalves that provided hardgrounds for the colonization by crinoids. These dm-scaled bioherms provided new habitats for brachiopods, serpulids, other bivalves and boring worm-like organisms.

The sudden increase in species richness of gastropods is another feature of the Middle Triassic main stage of marine recovery (Figs. 1, 2, 3 and 4, 5A,5B, 6B and 7) that was at least partly linked to the resurgence of large carbonate platforms. As shown by Friesenbichler et al. (2019) fig. 14, the ratio of gastropod versus bivalve species is much higher in carbonate platforms than in other settings. However, the data of Friesenbichler et al. (2019) might even underestimate the true signal. Roden et al. (2020) compared the diversity of the Cassian Formation, an exceptional Triassic lagerstätte, with the diversity of the Wetterstein Formation, which is environmentally comparable and approximately of the same age. They found that all mollusks were underrepresented in the Wetterstein Formation by factor 7 in comparison to the Cassian Formation, but gastropods by the astonishing factor 87. The cause for the affinity of gastropods to carbonate platform facies remains speculative, not at least because the life habit and feeding mode of gastropods cannot be deduced from the shell morphology. In spite of this limitation, we put forward the hypothesis that the quick diversification of gastropods in carbonate platform environments might have been related to the proliferation of dasycladacean algae, which were dominant carbonate producers in many settings where gastropods became diverse.

Palaeoecological studies have previously suggested that at least some Triassic gastropods lived in association with macroalgae (Fürsich & Wendt, 1977; Sellwood & Fürsich, 1981; Stiller, 2001; Hagdorn, 2004; Nützel & Schulbert, 2005; Diedrich, 2010; Urlichs, 2014). This conclusion was derived from the ecology of Recent macroalgae (e.g., Davies, 1970; Taylor, 1971; Thomassin, 1971; Brasier, 1975; Poulicek, 1985; Sánchez-Moyano et al., 2000; Chemello & Milazzo, 2002; Antoniadou & Chintiroglou, 2005; Pitacco et al., 2014; Duarte et al., 2015; Chiarore et al., 2017), on which gastropods grazed (e.g., Underwood, 1980; Johnson & Mann, 1986; Williams, 1993) and thereby enhance the photosynthetic capacity of the host algae (Amsler et al., 2015). The Triassic fossil record of non-calcifying algae is virtually non-existent, but the diversification and abundance of dasycladaceans at the beginning of the Middle Triassic is well established. In our dataset, only 5 out of 37 lithological units contain information about the number of Dasycladacea species associated with the investigated faunas (e.g., Salomon, 1895; Ogilvie Gordon, 1927; Schmidt, 1928; Schmidt, 1938; Granier & Grasović, 2000; Russo et al., 2000; Emmerich et al., 2005; Piros & Preto, 2008). Given the small number of datapoints, the positive correlation between the richness of dasycladaceans and gastropods is statistically not robust (r2 = 0.25; p = 0.32), but it is strengthened by an indirect line of evidence. Middle Triassic gastropod diversity decreased towards deeper water settings (Figs. 2 and 4), along with declining light. Because light limits the distribution of algae, it is certain that this trend was matched by decreasing macroalgae abundance.

Conclusions

Data on species richness from 37 Early and Middle Triassic lithological units containing level-bottom communities indicate that most of the analyzed taxa and guilds followed a hyperbolic-damped diversity trajectory, with an extended Early Triassic lag phase followed by an explosive increase in diversity at the beginning of the Middle Triassic that levelled out in the Ladinian. The Early Triassic delay in rediversification of benthic organisms has conventionally been attributed to ongoing environmental stress at that time, but the explosive (hyperbolic rather than exponential) diversity increase at the beginning of the Middle Triassic cannot be explained by relaxation of environmental stress. Our data therefore support a model in which the intensity of biotic interactions determines the rate of diversification. Accordingly, the Early Triassic delay in rediversification represents the time in which the intensity of biotic interactions was too low for driving a major diversification because of the dramatic loss of species during the end-Permian mass extinction. Conversely, the hyperbolic Middle Triassic diversity increase occurred when self-accelerating processes became effective, and it stopped when niche-crowding prevented further diversification. This pattern is in agreement with a competition-driven diversification model that predicts a reversal in the effects of interspecific competition on diversification rates from accelerating to damping. Apart from interspecific competition, other ecological feedbacks might have stimulated the hyperbolic increase in diversity during the Anisian. Carbonate platforms, which reappeared simultaneously with the main stage of benthic rediversification, provided the ecological context for rich hardground communities. The coincidence of the resurgence of carbonate platforms and the evolution of reef-builders with the main diversification of benthic faunas suggests a positive loop effect, which is confirmed by several case studies. When reefs diversified, other reef-builders became abundant and acted as ecological engineers and niche constructers, providing new habitats for additional species. The quick diversification of gastropods in Middle Triassic carbonate platforms might be related to the proliferation of macroalgae, which provided much improved opportunities for grazing. A strength of the biotic interaction model is that it correctly predicts both an extended lag phase and a hyperbolic increase in diversity thereafter, whereas environmental scenarios fail to predict the second aspect.

Supplemental Information

Supplemental Information 1 Supplemental Data

(A) List of lithological units used for this study including information about their country, age and faunal content. (B) species list including information about their occurrences, ecological guild, taxon and references. (C) Reference list.

Click here for additional data file.

EF is deeply grateful to Alexander Pohle for interesting and helpful discussions and his loving support. Christian Klug is thanked for helpful discussions. The paper benefitted from constructive reviews by William Foster, Alexander Nützel and an anonymous reviewer. Helpful inputs by editor Diogo Provente are gratefully acknowledged.

Additional Information and Declarations

Competing Interests

Author Contributions

Data Availability

The authors declare there are no competing interests.

Evelyn Friesenbichler performed the experiments, analyzed the data, prepared figures and/or tables, authored or reviewed drafts of the paper, and approved the final draft.

Michael Hautmann and Hugo Bucher conceived and designed the experiments, authored or reviewed drafts of the paper, and approved the final draft.

The following information was supplied regarding data availability:

The raw data are available in the Supplementary File.

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
