# Peer review of "The main stage of recovery after the end-Permian mass extinction: taxonomic rediversification and ecologic reorganization of marine level-bottom communities during the Middle Triassic"

_PeerJ, doi:10.7717/peerj.11654_

## Round 0.1 · original submission · Major Revisions

I have received now three comprehensive reviews about your manuscript. R1 noticed that there's currently a debate on the topic of the paper, but authors only included arguments from one side of the discussion. He also makes strong arguments about the quality of your data, pointing to key literature that are missing. You definetely should include arguments suggesting that the delayed recovery of the post-Permian is due to environmental controls. R2 also argue that your hability to discriminate between diversification models. As far as I understand, your hability to discriminate between "biotic control" and "environmental control" as drivers of diversification is limited without explicit phylogenetic data. I'm not sure if you can fit diversification models to data without phylogenies. I'm referring to the papers by Daniel Rabosky on ecological limits and density-dependent diversification (Rabosky 2013) and tools like RPANDA, DDD, and BAMM.
https://www.annualreviews.org/doi/abs/10.1146/annurev-ecolsys-110512-135800
I also agree that it'd be good to quantify functional redundancy, there's a recent review paper on it by Ricotta et al
https://besjournals.onlinelibrary.wiley.com/doi/full/10.1111/2041-210X.12604

https://www.sciencedirect.com/science/article/pii/S1470160X20304258?casa_token=pf4mD5SldUcAAAAA:eHmUhusJh8tYyU8YvZ-LTkC9b1vo3UoQSFXEI06Nr-qgjvA5YYCyjJWNfLtca-7OpHxabTtQOww

I do not think it's reliable to run a linear regression with only six data ponts. Please, remove this analysis. Pay closer attention to the comments by R1 and R2.

Introduction is really short, with only one paragraph. Please, also separate the Methods into subheadings for a better reading experience and allow reader to quickly find information. PeerJ also has a structured Abstract that I higly recommend you to follow. Discussion is quite long for the amount of results you have, with 5 pages. Try to reduce it somehow.

·

Basic reporting

This article meets the basic reporting requirements, but I do feel that the background/context could be improved (see review)

Experimental design

This article meets the experimental design requirements, but I do think some data is missing but the impact of the missing data may be insignificant.

Validity of the findings

This article meets the validity of the findings requirements, but I do think some ideas from the other side of the debate have been overlooked.

Additional comments

The Middle Triassic is most likely exceptionally more diverse then we realise! I think the work of Friesenbichler et al. in uncovering this data is great and I really support more of it. This is a nice study that looks at a tightly vetted dataset of benthic marine life and provides new insights into the patterns of changes in both taxonomic and ecological diversity following the end-Permian mass extinction. There is a lot to like about this manuscript, and it clearly represents an important contribution to the literature. The figures are great, and the overall quality of the English is high! I do also think there are requirements for revisions to how the manuscript is currently structured/focussed. My comments are below:

Dataset
The dataset looks good I mostly like the justification for excluding a lot of the literature (another way you could had worded it, is that you are only using fossil occurrences that are supported by material that has been accessioned). But I do have two concerns/comments:

1. I don’t believe your dataset is complete, for example the benthic fauna described from the Great Bank of Guizhou [Foster et al. 2019, Papers in Palaeontology 5, 613-656] which included a full systematic paleontology. Other examples include data from the Aggtelek reef (e.g., by Hand Hagdorn), and data from Svalbard published by Hans Frebold in the early 1900s.

2. You are excluding identifications based on thin sections and polished slabs. Okay I can understand why you would do this for groups like bivalves and gastropods, but for other groups like foraminifera, sponges, bryozoans that rely on thin sections for accurate taxonomy then how can you justify their exclusion? This may be why a lot of the Aggtelek reef data is missing (as an example).
Overall “Competition in slow motion” argument
I found the introduction/discussion quite narrow-minded and misleading. The debate on the delayed recovery has four aspects: (1) competition in slow motion (2) persistent environmental stress (3) subsequent biotic crises (4) poor fossil record means we underestimate the real biological patterns. I agree that this paper very briefly mentions points 2-4 but it really pushes its own agenda of point 1.

I have a number of issues with this:

1. The abundant data that serves as independent proxies (inorganic geochemistry) for environmental conditions that are detrimental to metazoans were pervasive throughout the Early Triassic, and that these conditions meant that there were a lot of lithological units that represent habitats that cannot support diverse ecosystems. If you take this into account, then you would expect a recovery that looks like this quick sketch below – and that is very similar to your results. None of this is really considered/falsified in this article.

2. This article collected data on the ecological information that the benthos gives us, but then doesn’t consider this in the debate. What are the ecological changes telling you? There are a number of ecological changes that point to environmental stress, rather than species-species interactions. For example, the dominance of infaunal deposit feeders, the increased relative size of a lingulid brachiopod lophophore (see Posenato et al., 2014 or Foster et al., 2018 for an alternative interpretation), the recovery of sessile suspension feeders, the high abundance of Lazarus taxa, the presence of “paper pectens”, small body sizes (ok the control on this is more controversial, but you still overlooked it), the replacement of microbial-dominated reefs by metazoan-dominated reefs, the high abundance of opportunistic taxa, and so on. I would expect that if functional redundancy correlates with total species diversity then this would support the role of species-species interactions in driving diversification rates rather than an environmental control – but again you overlook the functional diversity data in your discussion – how do your functional diversity results differ to the Foster and Twitchett 2014 study for example (but also the study by Dineen et al., 2014 and many other scientists) and what implications does that have?

Likewise, the ecological changes are consistent with the Mesozoic oceanic anoxic events, which would point to an environmental control rather than a pure biological control as this manuscript argues.
In addition, Foster et al. (2017) dedicated some thought to the idea that the “Unionites” are actually something else with a very different feeding ecology and this has a big impact on our interpretations of recovery. But there is no mention of these issues in this manuscript, not just for “Unionites” but also gastropod ecological interpretations, they appear to be overlooked.

3. Citations, it is noticeable that there is a preference/exclusive referencing of scientists associated with your research group only! Firstly, this is not very scientific to just ignore the literature that is already out there, and secondly why would these other research groups read/cite this article if their decades of research are overlooked? It is disheartening to say the least.

4. If you’re interested, in my view the delayed recovery is a combination of all four parts of the debate, which is why it is so hard to separate them and why you should discuss all of them!

Other comments:
I think a lot would be gained by looking at the functional redundancy methods of Catalina Pimiento (UZH & Swansea University) and seeing how you can implement those given that you have all this data.

Line 79: change “Americas” to “USA”, you have no data from Canada, Mexico, Latin America or South America.

Lines 221-238: It seems odd that you do not cite Hofmann (2019) PNAS 2019 116 (1) 79-83, given the large overlap.

Line 261: “scarce data” actually this may be a misinterpretation. For example there are thousands upon thousands of Anisian brachiopods in the European Muschelkalk, but they represent very few species. So the data is not scarce, but instead some groups suffered such severe bottlenecks that diversification rates were still really low in the Middle Triassic. What’s more, ecologically, the articulate brachiopods loose a large number of ecological guilds and only hard substrate ecologies occur after the extinction (see VÖRÖS ,et al. Rivista Italiana di Paleontologia e Stratigrafia vol. 125(3): 711-724).

Line 272-274: Foster et al. (2017) [Svalbard data] did not report everything. The Svalbard sections are much more diverse than your dataset suggests/ what we’ve published so far. I.e., a sampling bias is overprinting the diversity of Svalbard.

Line 291-295: “Most general statements on the biotic recovery that followed the end-Permian mass extinction ignored this key point.” This is just not true, everyone talks about this but not everyone always mentions it in publications.

“Such a decoupling is not compatible with the exclusive role of abiotic stressors of global scale (sea-surface temperature, sea level changes, oxygenation of the water column, pH, salinity, etc.)” again not true, or I do not understand your logic. This decoupling is consistent with every OAE. E.g., an anoxic seafloor but oxygenated surface waters would explain this signal. It would be great if you could expand your reasoning here.

Lines 324-341: see Martindale, et al. Palaeogeography, Palaeoclimatology, Palaeoecology 513 (2019): 100-115. The thing is, just as the Proterozoic microbial buildups represent reef ecosystems, so do the Early Triassic microbial buildups. On top of that, Friesenblichler et al. (2017) + Heindel et al (2018) demonstrated that these reefs are microbial-sponge reefs and are much larger and more expansive than the younger Brayard et al. transient reefs (see also the recent paper I wrote in the Depositional Record). On top of that again, Zatoń et al. also reported microconchid buildups from the Griesbachian. None of these, including the USA biostromes, are comparable to the Middle Triassic platform margin reefs, which I think is the point. Also note, Gliwa et al. (2020) Fossil Record 23, no. 1: 33-69. report a 2km wide deposit of keratose sponge spicules immediately at the end-Permian event. This would be the oldest post-extinction sponge biostrome.

Line 328: Why do you cite Popa et al. only? They suggested this following the work already published by Flügel (2002) and Payne et al. (2006) and many others. Likewise, why focus on the Romanian reef? I assume it is because you know it best, but the reef that really represents the first research and oldest platform margin reef is the GBG reefs studied by Payne et al. (2006) amongst others (for example Dan Lehrmann has spent >30 years studying those limestones).

Line 353-375: One interesting biological change associated with molecular fossils is the switch from assemblages dominated by hopanoids in the Early Triassic to one dominated by sterols in the Middle Triassic. This suggests a switch in the main primary producers in the oceans from bacteria-dominated in the Early Triassic to eukaryote(algal)-dominated in the Middle Triassic, which also coincides with your gastropod recovery signal. This would have caused differences in the evolution of gastropod ecologies, i.e., bacteria-grazing in the Early Triassic and algal-grazing in the Middle Triassic – but again this relationship between environment and evolution [to me] seems largely overlooked.

Line 397: “stabilization of the carbon isotope curve” so this is based on Payne et al. (2004) Science but actually if you look at the Middle Triassic carbon isotope curves from the European sections they show large fluctuations.

Fig. 8. A linear regression between 6 data points is meaningless.

I hope you find these comments constructive and I’m available to the authors if they have any questions,

Reviewer 2 ·

Basic reporting

In their ms, Evelyn Friesenbichler and co-authors analyze the Triassic re-diversification of benthic fauna after the P/T extinction and explore the respective, putative roles played by abiotic and biotic ecological factors to explain the delay in the rediversification of the benthos compared to the nekton. Using species richness data from benthic communities from 35 Early and Middle Triassic lithological units they show that most taxa follow a hyperbolic-damped curve diversity model with a rapid and simultaneous diversification in the early middle Anisian. The authors discussed the relevance of different hypotheses and scenarios stressing the importance of either abiotic and biotic factors to explain the 4 My delay in the re-diversification of the benthos after the P/T crisis. They hypothesize the hyperbolic-damped curve model supports the relevance of biotic interactions as driving factors of the diversification (competition-driven diversification model). They also hypothesize that the decoupling in the post P/T recovery between the nekton and the benthos is due to different competition intensities and habitat homogeneities. Finally, they emphasize the importance of carbonate platforms and reef expansion to promoting benthic rediversification and community richness.

The ms reads well and references are relevant to the topic. In the form, the ms structure, figures, and tables are appropriate, informative and detailed, raw data are provided as supplementaries.

Experimental design

Even if not novel, understanding the diversification of marine life after the P/T crisis remains a highly topical and mostly unresolved issue in macroecology and macroevolution. The authors address it using both ecological inferences and conceptual diversification models to disentangle the respective roles of environmental drivers. The analysis relies on an important dataset.

Such a macroecological approach to unreveal major diversification patterns inevitably implies a simplification of ecological processes and species life traits, guessing that biodiversity patterns are well-marked enough to be highlighted. However, in the present work, the interpretation of ecological traits is oversimplified as entire taxonomic classes are assigned to one single trophic class, which is sometimes very unlikely and partly erroneous (bivalves excepted as unexpectedly, the ecology of this class is well detailed). This oversimplification of the ecological part also applies to the interpretation of the correlation between gastropods (which are all considered herbivore) and macroalgae, which suffer from a very poor fossil record. This oversimplification constitutes one major issue of the analysis and of the intepretation of results. Besides, while paleoecology is a main focus of the study, there is no mention of the analyzed ecological guilds in the conclusion. This is unexpected. Methodology, sampling strategies and their justifications constitute a second major issue of the study. In particular, the M&M section suffers from several shortcomings and circular reasonings (see below). Mainly, many samples were omitted and the adopted methodology is justified by very questionable arguments. Finally, there is no statistical test to justify the regression models.

Detailed comments on Material &Methods

lines 71-72: " Benthic communities containing less than five species were considered as undersampled and not representative and were therefore not considered."
Even if this strategy may partially limit the bias of synonymies and taxonomic redundances, it may also lead to underestimate diversity of the fossil record, which is far from being well known. In addition, this may also limit the sampling of lithological units, thereby of certain habitats, environnments and associated faunas, especially when few, specialized species are strictly associated to peculiar environments (ex : dysoxic marine environments)

lines 78-80: "Our approach led to a certain overemphasis of European lithological units (26) in comparison to 79 those from Asia (four) and the Americas (five), but the prevailing cosmopolitism of marine 80 faunas of this time ...". Cosmpolitanism of the fauna is a result and remains to be demonstrated. This cannot be used as a methodological argument beforehand to defien the methodology. This can be further discussed in the discussion section, but not in the M&M.

lines 81-83: "Because environmental conditions did not have significant effects on the diversification of the benthos (see Discussion) we did not consider the palaeogeographic position of the investigated lithological units". This sentence is very confusing because palaeogeography and environments are distinct issues, which are not necessarily related. In addition, as paleoecology is the main focus of the study, you cannot argue that environment does not matter ! Please can you clarify the sentence. In addition, results cannot be used as an argument to justify the method as this may read as circular reasoning ! I think this sentence needs to be rephrased and clarified, or even omitted. Here again, the palaeogeographic bias is, unfortunately common to most large-scale paleoecological studies and can be discussed as a possible, but inescapable limitation in the discussion section. Paleontologists can understand and admit it.

107-109 : "ophiuroids were collectively assigned to ... epifaunal detrivore-suspension-feeders, because evidence for more specialized feeding is lacking."
Simplification of fossils' life traits cannot be avoided in large-scale paleobiological studies, the guess being that evolutionary patterns are clear enough to be evidenced. However, I have serious doubts here as to the classification of opihuroids as just "epifaunal detrivore-suspension-feeders". I am afraid this is by far oversimplified. Most opiuroids are not suspension feeders, many are carnivorous for instance. This also applies to the classification of all gastropods and regular echinoids as "epifaunal herbivore and/or detritus-feeders". Gastropod and echinoid diversification in the context of the Mesozoic Marine Revolution also relies on the diversification of feeding habits including carnivorous strategies, especially in "cidaroids". I think this is oversimplification and partly incorrect.

Line 118-120 : "Unfortunately, compiling reliable global species level curves is virtually impossible because ....of the sheer amount of working time required for this". I would recommand to omit such an argument; the necessary amount of work can not be used as an argument to justify a methodological bias.

124-125 : "... for each substage and ecological guild, the lithological unit that contains the
highest number of species was used as a reference point." Using such an approach to avoid synonymy and preservation biases makes sense but it may also lead to a strong bias and to underestimating diversity and the complexity of habitats.

127-129 : "Also differences in preservation play no major role in this approach, because it can be assumed that lithological units with exceptionally high diversity were little affected by significant preservation problems." If this was really the case, this must rely on serious taphonomic evidences. Even preservation lagerstätten are affected by preservation bias ! Are all the lithologicla units you worked on are recognized lagerstätten. I think this is very questionable.

132-134 : "Synonyms and open nomenclature are not a problem here either, because there is no reason to assume that there is a systematic bias between taxa in the amount of taxonomic uncertainty." This reads as a gratuitous and unnecessary comment. Do you have any element to support it ?

Validity of the findings

Detailed comments on Discussion

240-242 : "Our data provide no support for the exponential model because most investigated taxa and guilds show evidence for dampening by the end of the Middle Triassic" Did you test it statistically ? As this may be due to the heterogeneity of data and sampling bias.

247-248 : "The majority of the semilogarithmic plots presented in this study (Figs. 5 and 7) conform to the hyperbolic-damped model". Is there any mathematical evidence for this ? This impossible to say from figs 5 and 7 only ! A regression model is needed here.

275-276 : " the history of research and the size of exposure influence our knowledge on species richness." Of course ! This is a serious limitation to diversity studies

301-305 : "Accordingly, the much precocious and explosive recovery of ammonoids in comparison to bivalves essentially reflects the higher intensity of interspecific competition among secondary consumers in comparison to primary consumers (Hautmann et al.,
304 2015). The present study focuses on marine level-bottom communities that are in general characterized by very low levels of interspecific competition (Stanley, 2008)." Despite the fact that saying that competition is more harsh in between upper consumers is a simplication of real life, I can admit this difference in competition pressure between trophic levels. However, there is also many carnivores at the sea botoom and there is no reason to assume that competition is lower at the sea bottom than in the water column ! For instance, are there less competition in coral reefs than in the open ocean ? I am not sure at all ! This time lag between the nekton and the benthos in diversification rates in the Triassic reminds of diversification patterns already noticed for other time periods (the Global Ordovician Biodiversification for instance). Usually, this is explained by the diversification of phytoplankton first, then Nekton diversified as it directly feeds upon it, and the benthos comes afterwards as a result of new food source availability near or upon the sea bottom and the evolution of new ecological niches. Such a scenario makes more sense here than the "competition " scenario. At least, the evolution of the nekton depends on phytoplankton, which should be discussed here.

In contrast, I totally agree with the importance of reef buildups as a factor promoting speciation (and evolution of new ecological niches also because complexity of biotic interactions increased, not only the complexity of microhabitats, even in the benthos !).

Finally, I am not convinced at all by the direct correlation between gastropod and macroalgal diversity. First because I am not sure gastropods used to feed directly upon macroalgae and second, the fossil record of macroalgae is extremely poor if there is one, as the authors recognize (line 365). There must be a third factor (habitat, depth, ... as the authors suggest) explaining this correlation (common habitats do not mean that all their inhabitants necessarily used to feed upon each other). I agree that the hypothesis can be formulated, but I would suggest to be more cautious as to the trophic interpretation that may rely on particular platform units, and I would suggest to remove the correlation that does not make much sense to me considering the nature of data.

Detailed comments on Conclusion

The conclusion reads unappropriate and in particular, the end of the conclusion is not really in line with the discussion and presents gratuitous and unnecessary statements, which are not related to the objectives (see my detailed comments below).

424-425 : "The dominance of epifaunal herbivores and/or detritus-feeders in middle Triassic shallow marine environments and carbonate platforms might be related to the proliferation of macroalgae". I totally disagree here. As the authors mention in the discussion, the quality of the fossil record does not allow for such a statement, without mentioning the very hypothetical herbivory of gastropods. Surprinsingly, here the conclusion is not in line with the very critical position of authors in the discussion section.

432-434 :"We propose a reverse scenario in which biologically-driven rediversification of the benthos contributed to the stabilization of the carbon budget through binding of carbon in large carbonate platforms via benthic calcifying organisms." This hypothesis is totally gratuitous and is not supported by the data and analyses presented in the study !

Additional comments

To conclude, I think there are many issues to address before this work can be published. This is very unfortunate because the subject is very topical, the analysis challenging and very novel results could have been expected for this intriguing time period! I really encourage the authors to amend their ms accordingly. Most issues are more in the form, the writing and the interpretation of data than in the analysis itself. I hope a revised version can be submitted in a near future.

·

Basic reporting

This is a very interesting paper that does not need much change - there are some issues listed below. In the light of the importance of gastropods in this paper you could alo consider Roden et al. 2020, Palaeontology for the role taphonomy may play in the abundance of gastropods.

Experimental design

Methods and questions clearly stated, simple but meaningful statistics; it is good that data were acquired directly from primary literature and own previous work

Validity of the findings

The graphs show very clear signals that are discussed in a reasonable manner. It is a little too obvious that the authors want recovery to be purely biologically driven, alternative abiotic drivers are mentioned but not taken really serious - at least in my impression.

Additional comments

44. cite also Nützel 2005 who showed this also in terms of standing diversity in gastropods

47/48. cite Senowbari et al.: Anisian (Middle Triassic) buildups of the Northern Dolomites (Italy): the recovery of reef communities after the Permian/Triassic crisis, 1993Facies 28(1):181-256
and see also discussion in Nützel 2005, p. 511/512; there is also more lit for instance Kiesslings reefs papers, the Flügel reef book


62-69, PBDB was not used, explain why

69, “ Taxa reported from thin sections or polished slabs were not considered because taxonomic assignments of these taxa were in most cases difficult to verify. “ this is a crude statesmen because on the other hand many benthic organisms (sponges, corals, algae, microbes etc) are best studied in thin section are

71: “Benthic communities containing less than five species were considered as undersampled and not representative … ” this sounds a little arbitrary, why 5 and not 4, 6, 7, …? I know eTr. assemblages with many reasonably preserved specimens but only two species (Werfenella Natiria from Werfen) and in my opinion this preserves a biological signal.

109: Insert, “However we are aware that this might be an over-generalization. “ or smth similar (many recent gastropod are parasites, carnivorous grazers or microcarnipovorous

174 onward, please provide one or two examples for each guild for instance “pedunculate suspension-feeders (e.g. articulate brachiopods)

for instance few people know an example for “Cemented microcarnivores”

293 this, by the way, applies also to the so-called lilliput effect that is absent in ammonoids, which may be pretty large even in the earliestr Triassic

303 a statement what ammonoids fed on is needed

326 much work was done before Brayard: Senowbari, Flüget etc

341 in this context it is noteworthy that the richest Muschelkalk gatropod fauna comes ftom the Diploporen Dolomit of Silesia, which it a calcareous rock with abundant dasycladcean alge

356 at least the Fürsich & Wendt study is speculative in this respect since algae were not substantiated in the association in question.

375 the algae-gastropod positive correlation is an association of phenomena, it might represent causality but could also reflect an underlying reason responsible for both, for instance change of abiotic factors that helped both, algae and gastropods; I would at least discuss this possibility

---

## Round 0.2 · accepted · Accept

Dear authors,

Thank you for your careful attention to the previous round of reviews. Based on comments from the original three reviewers, I would like to accept this paper and move it forward.

However, please note that there are a few minor changes that need to be made prior to publication (see the reviewer comments).

Best,

Brandon P. Hedrick (Ph.D.)

·

Basic reporting

NA

Experimental design

NA

Validity of the findings

NA

Additional comments

The resubmission by Friesenbichler et al. does well to address the many comments from the initial review and continues to represent a novel and interesting investigation on the diversification of benthic marine ecosystems in the Middle Triassic. There are still aspects I disagree with, but the arguments and data are both of high-quality and I commend the authors for such an interesting manuscript – especially the part about the role of carbonate platforms. I think some of the rebuttal comments were intentionally provocative (e.g., I did not say Unionites is a nuculid, instead I suggested it represents a dustbin taxon that doesn’t only represents suspension feeders) but I do not feel the need to respond to them here. Also note, that Hans Frebold’s fossils are mapped so well that you can find the horizons in the field still today almost 100 years later but I don’t think these omissions affect the quality of the manuscript and findings, and the methods are clear to explain the absence of such occurrences.

Two very minor comments:
line 89-91, change (nine species) to (9 species) and so on for the rest of the list.

line 91-93. See the provocative comment by Richard Twitchett on the Discussion on Lazarus taxa and fossil abundance at time of biotic crisis. Journal of the Geological Society, London, 157, 511-512. My point here, is that Asia has fewer rock unit names because there are fewer geopolitics that leads to multiple names for the same rock formation.

Reviewer 2 ·

Basic reporting

In the new version of their ms, Evelyn Friesenbichler and co-authors carefully adressed, with details all my concerns on a first version of this ms. They justified their position and methodological choice, and/or took into account my remarks and modified the text accordingly. Overall, I find the new version clearly gained in clarity, is now better documented, statements are more clearly supported and explained including the analytical and methodological strategy.
In particular, the quality of the discussion and conclusion have been significantly improved. The conclusion is more clearly linked to the original research question and analytical results.
The over-generalization in the interpretation of taxon ecology is better explained and justified including the relation between gastropod and algae diversity, which is better detailed and supported by references.

Experimental design

Overall, I find the new version clearly gained in clarity, is now better documented, statements are more clearly supported and explained including the analytical and methodological strategy.

Validity of the findings

no comment.

Additional comments

In the new version of their ms, Evelyn Friesenbichler and co-authors carefully adressed, with details all my concerns on a first version of this ms. They justified their position and methodological choice, and/or took into account my remarks and modified the text accordingly. Overall, I find the new version clearly gained in clarity, is now better documented, statements are more clearly supported and explained including the analytical and methodological strategy.
In particular, the quality of the discussion and conclusion have been significantly improved. The conclusion is more clearly linked to the original research question and analytical results.
The over-generalization in the interpretation of taxon ecology is better explained and justified including the relation between gastropod and algae diversity, which is better detailed and supported by references.

I have no more comments at this stage. I think this is a very topical and valuable work that is now ready in its present form to be published as it is.

·

Basic reporting

The authors put quite some work in this manuscript and have addressed the points of the previous reviews suffiecintly; English is good, structure, figures are, context etc. are all good.

Experimental design

no comment

Validity of the findings

no comment

Additional comments

Line 273/274; when you mention possible abiotic factors, please mention at least the most important ones that were discussed in the literature, such as high temperatures, unstable carbon cycle as evidenced by Carbon isotope fluctuations, anoxia ....